# Robust Management of Systemic Risks and Food-Water-Energy-Environmental Security: Two-Stage Strategic-Adaptive GLOBIOM Model

Tatiana Ermolieva *, Petr Havlik, Yuri Ermoliev, Nikolay Khabarov and Michael Obersteiner

International Institute for Applied Systems Analysis, Laxenburg 2361, Austria; havlikpt@iiasa.ac.at (P.H.); ermoliev@iiasa.ac.at (Y.E.); khabarov@iiasa.ac.at (N.K.); oberstei@iiasa.ac.at (M.O.)
* Correspondence: ermol@iiasa.ac.at; Tel.: +43-2236-807581

**Abstract:** Critical imbalances and threshold exceedances can trigger a disruption in a network of interdependent systems. An insignificant-at-first-glance shock can induce systemic risks with cascading catastrophic impacts. Systemic risks challenge traditional risk assessment and management approaches. These risks are shaped by systemic interactions, risk exposures, and decisions of various agents. The paper discusses the need for the two-stage stochastic optimization (STO) approach that enables the design of a robust portfolio of precautionary strategic and operational adaptive decisions that makes the interdependent systems flexible and robust with respect to risks of all kinds. We established a connection between the robust quantile-based non-smooth estimation problem in statistics and the two-stage non-smooth STO problem of robust strategic–adaptive decision-making. The coexistence of complementary strategic and adaptive decisions induces systemic risk aversion in the form of Value-at-Risk (VaR) quantile-based risk constraints. The two-stage robust decision-making is implemented into a large-scale Global Biosphere Management (GLOBIOM) model, showing that robust management of systemic risks can be addressed by solving a system of probabilistic security equations. Selected numerical results emphasize that a robust combination of interdependent strategic and adaptive solutions presents qualitatively new policy recommendations, if compared to a traditional scenario-by-scenario decision-making analysis.

**Keywords:** interdependent systems; systemic risks; two-stage non-smooth stochastic optimization; quantile-based risk constraints; robust strategic and adaptive decisions



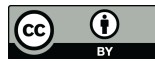

## 1. Introduction

Increasing interdependencies among systems involving interactions between man, nature, and technology resemble a complex chain network connected through supply–demand relations. Disruption of such networks can trigger systemic risks associated with critical imbalances and exceedances of vital thresholds, affecting food, energy, water, and environment (FEWE) security at various levels, with possible global spillovers. Risks of disruptions and failures in such systems may be unlike anything that has been experienced in the past. These risks can be induced by human decisions, in combination with natural shocks. For example, an extra load in a power grid triggered by a power plant or a transmission line failure can cause cascading failures with catastrophic systemic outages [1]. In financial networks, an event at a company level can lead to severe instability or even to a crisis similar to the global financial crisis 2008. A hurricane, in combination with inappropriate dams' maintenance and land-use management, can result in human and economic losses, similar to those induced by Hurricane Katrina [2]. Another example relates to an increase of biofuels production, which affects crops and food prices, destabilizes food and water provision, and worsens environmental conditions [3,4]. The notion of systemic risks was introduced by Kaufman and Scott [5] in relation to financial systems. The definition has been adopted for other natural and anthropogenic systems, e.g., power

grids and critical infrastructures [6], biodiversity [7], financial and insurance systems [8–10], and other natural and anthropogenic systems [11,12].

Systemic risks in interdependent FEWE systems can be defined as the risks of a subsystem (a part of the system) threatening the sustainable performance and achievement of FEWE security goals. Thus, a shock in a peripheral subsystem that is induced (intentionally or unintentionally) by an endogenous or exogenous event can trigger systemic risks propagation with impacts, i.e., instability or even a collapse, at various levels. The risks may have quite different policy-driven dependent spatial and temporal patterns. While standard risks analysis and assessment can rely on historical data, systemic cascading risks in FEWE systems are implicitly defined by the whole structure and the interactions among the systems, in particular costs, production and processing technologies, prices, trade flows, risk exposure, FEWE security constraints, risk measures, and decisions of agents.

Prediction of systemic risks in integrated natural and anthropogenic policy-driven FEWE systems is a rather tedious task. The main issue in this case is robust management of the risks, which can be achieved by equipping the systems with precautionary and adaptive strategies enabling the systems sufficient flexibility and robustness to maintain sustainable performance and fulfill joint FEWE security goals independently of what systemic shock occurs [13,14]. In front of uncertainties, the strategies (decisions) can be of the two main types: the ex-ante strategic precautionary anticipative actions (engineering design, policy setting, resource allocation, technological investments, and water and grain reserves) and the ex-post adaptive adjustments (marketing, inventory control, subsidies, prices, and costs) that are made after making observations and receiving additional information about uncertain system parameters. The anticipative and adaptive measures can reduce the tightness in various supply–demand relations [15] and lessen chances of critical imbalances, exceedances of vital thresholds, which could otherwise lead to systemic failures and the lack of FEWE security.

A portfolio of robust interdependent ex-ante and ex-post strategies can be designed by using a two-stage stochastic optimization (STO) approach incorporating both types of decisions (see, e.g., References [16,17]). The two-stage STO has been applied in studies for dealing with systemic interdependent risks, e.g., for integrated catastrophic risk management [18–23], for agricultural risks management [24–26], for energy security management [27–30], for robust operation of multipurpose reservoirs [31], and for climate change risk analysis [32]. In this paper, we discuss the implementation of the two-stage STO approach within a multi-regional, multi-sectorial recursive-dynamic, partial-equilibrium Global Biosphere Management (GLOBIOM) model for managing systemic risks and FEWE security in land-use systems (see Section 3 and References [33,34]). The model includes complex supply–demand chains established between regions, sectors, producers, and consumers, requiring a proper attention to the emergence and management of the systemic imbalances. In the case of a single region (or sector) model, as it is discussed in Section 2, the supply–demand imbalances and the risks are connected with possible costs for treating two situations: dealing with shortage and excess of production. When there is more than one region or sector connected through joint supply–demand relations and resource constraints, the systemic risks of imbalances and the associated costs dependent on the parameters characterizing both regions (sectors), e.g., region-specific risk exposure, production potentials, investments, operational costs, and feasible strategic (precautionary) and adaptive (operational) decisions. The problem of managing systemic risks in a multi-regional and multi-sectorial model GLOBIOM is a much more challenging task. An example of an agricultural supply chain in GLOBIOM involves, e.g., the following three main stages: crop production, from production to processing plant (different processing technologies), and from the plant to clients (regions, sectors, producers, and consumers), in the form of crops, biomass, and crop- and forest-related biofuels. Systemic risks can be triggered by stochastic crop yield (the model may also incorporate stochastic costs, resource, e.g., water availability or/and requirements, etc.) shocks. Section 3.1 argues that managing systemic risks in FEWE systems requires the mobilization of the existing

and the creation of additional strategic and adaptive activities and measures to equalize supply and demand in complex supply–demand chains at various spatiotemporal levels. The imbalances are represented by means of expected surpluses and shortfalls (deficits) defining the systemic risks and characterizing the demand for additional strategic and operational measures to reduce or eliminate the imbalances. Management of systemic imbalances can be addressed by solving a system of probabilistic security equations as it is discussed in this paper (see Sections 2–4).

The paper has the following structure. Section 2 discusses the notion of robustness in statistics and establishes a connection between the robust quantile-based non-smooth estimation problem in statistics and the two-stage non-smooth stochastic optimization (STO) for robust strategic–adaptive decision-making in the presence of uncertainty. The term "robust" was introduced in statistics [35] to identify statistical methods, which are not unduly affected by "bad" observations or outliers. The mean is not robust to outlier, whereas the median or quantile-based estimate is robust [36–38]. The robust quantile-based estimation problem is closely related to stochastic minimax problems of FEWE security analysis and two-stage strategic–adaptive systemic risk management. In the two-stage STO problems, the risk aversion emerges due to the coexistence of ex-ante strategic and ex-post adaptive decisions in the form of Value-at-Risk (VaR) quantile-based risk constraint [13,14,18–23]. A basic two-stage production planning model with the two types of decisions, strategic and adaptive, to minimize the risks of systemic imbalances in supply–demand relations is discussed in Section 2.2. The strategic ex-ante scenario-independent first-stage decisions are taken before uncertainty is resolved and operational ex-post scenario-specific second-stage decisions are made after the values of stochastic parameters become available, i.e., after learning additional information. It is shown that the adaptive second-stage decisions depend non-smoothly (because of max or min operators) on the first stage strategic decisions (path-dependencies) and on the uncertainty scenarios, providing strong cross-period random interactions among decisions. Similar models are relevant for planning energy (bio-based for heat and power generation) capacity investments under stochastic biomass demand, for planning agricultural technologies (e.g., irrigation or storage capacities) or hydropower-production-capacity expansion under stochastic water demand [39], and for many other situations (see References [24–31] and references therein).

The idea of the robust two-stage strategic–adaptive planning is incorporated into a global land-use model GLOBIOM, showing that management of systemic risks in a large-scale, multi-regional, and multi-sectorial model can be addressed by solving a system of probabilistic security equations. A two-regional (two-sectorial) fragment of GLOBIOM in Section 3.1 demonstrates that systemic risks arise as a result of joint resource and security constraints, risks exposures, and decisions of various agents. The general two-stage strategic–adaptive GLOBIOM is formulated in Section 3.2. The model suggests a robust combination of complementary strategic and adaptive decisions. Section 4 discusses selected numerical results. In particular, robust grain storage is an important measure to relax tight supply–demand relations, enabling systemic flexibility, smoothing production imbalances, and meeting security goals at regional and global levels. Moreover, robust storage (grain and water reserves) allows to avoid unwishful land expansion and costly investments, e.g., in additional irrigation technologies. Section 5 summarizes the main conclusions.

## 2. Robust Decision Analysis: Strategic and Adaptive Decisions

### 2.1. Robustness Concept

In order to use the notion of "robust decisions" for systemic risk and security management in interdependent systems, this section briefly discusses the concept of "robustness" in statistics and general decision-making.

### 2.1.1. Robustness in Statistics

The term "robust" was introduced in statistics [35] to identify statistical methods, which are not unduly affected by "bad" observations or outliers. In the presence of outliers, classical estimation methods, such as least square analysis, regression, and variance/covariance analysis, may have poor performance. In the case of Big Datasets, analysis of outliers is important because their rejection may delete important and relevant observations. Huber [35] defined robustness based on a probabilistic minimax approach optimizing the worst that can happen in some probabilistic sense. Therefore, robust statistical analysis and estimation problems rely on non-smooth minimax stochastic optimization principles. Optimal solution minimizing the quadratic function is the mean value of a random variable, $\omega$, whereas the median—and more generally a quantile—minimizes function:

$$M(x) = E(x - \omega)^2 = \int (x - \omega)^2 P(d\omega) \tag{1}$$

$$Q(x) = Emax\{\alpha(x - \omega), \beta(\omega - x)\} = \int max\{\alpha(x - \omega), \beta(\omega - x)\} P(d\omega) \tag{2}$$

where $max\{\alpha(x - \omega), \beta(\omega - x)\}$ is a non-smooth random function, and $x$ belongs to a set of feasible decisions, $X$ and $x \in X$. The probability space $(\Omega, F, P)$ is defined by the stochastic variables, $\omega$ and $\omega \in \Omega$; the event space, $F$; and the probability measure or probability distribution, $P(d\omega)$. Parameters $\alpha$, $\beta > 0$ are costs (penalties) associated with over- or under-estimation, $\omega$. In general, the costs (cost functions) can be stochastic and depend, for example, on $\omega$. Minimization of (2) is a stochastic minimax problem. Both functions $M(x)$ and $Q(x)$ are convex. Let us assume that $Q(x)$ is a continuously differentiable function, i.e., $P$ has a continuous density. If so, then we get the following:

$$Q'(x) = \alpha P[\omega < x] - \beta P[\omega \geq x] = 0 \tag{3}$$

Moreover, the solution, $x_q$, of the problem (2) satisfies the following equation:

$$P[\omega \geq x] = q, \ q = \frac{\alpha}{\alpha + \beta}.$$

For $\alpha = \beta$ Equation (3) defines the median. Solution $x_q$ minimizes also function

$$x + (1/q)Emax\{0, \omega - x\}, \ q = \frac{\alpha}{\alpha + \beta}, \tag{4}$$

which follows from the equivalent rearrangement of function $Q(x)$ in the form $Q(x) = \alpha x + (\alpha + \beta)Emax\{0, \omega - x\} - \alpha E\omega$. Model (2) is an important stochastic minimax problem used in the security analysis and two-stage strategic–adaptive risk management (Section 2.2). According to (3,4), the robust solution, $x_q$, is characterized by quantiles and costs $\alpha$ and $\beta$, which is also discussed in Sections 2.2 and 3. In the case when $Q(x)$ is not continuously differentiable and $P$ does not have a continuous density function, optimality condition (3) has to include subgradients [40–42] of function (2). Within a set of solutions of Equation (3), the robust decision, $x_q$, is defined as the minimal $x$ satisfying equation $P[\omega \geq x] \leq q$.

### 2.1.2. Robust Decision-Making under Uncertainty

For the analysis of complex systems, various performance indicators, $f_i(x, \omega)$ and $i = \overline{0, m}$, can be used (see, e.g., References [18–32,43–45]) to evaluate robustness of decisions $x$. For example, if $x$ defines production by different crop production systems, then $f_i(x, \omega)$ can indicate the level of water/air/soil pollution or food calories per person for each combination of feasible production level, $x$, and plausible $\omega$. The problem of robust decision-making is to find such production, $x$, that the goal function $f_0(x, \omega)$ (net profits or costs) is maximized (or minimized) and the indicators, $f_i(x, \omega)$ and $i = \overline{0, m}$, identifying required norms, e.g., environmental or food security, are fulfilled for all scenarios, $\omega$. This

leads to an STO model formulated as optimization (maximization or minimization) of an expectation function:

$$F_0(x) = Ef_0(x,\omega) = \int_\Omega f_0(x,\omega)P(d\omega) \tag{5}$$

It is subject to the following constraints:

$$F_i(x) = Ef_i(x,\omega) = \int_\Omega f_i(x,\omega)P(d\omega) \geq 0, \; i = \overline{0,m} \tag{6}$$

where $x \in X \subseteq R^n$ and $\omega \in \Omega$ represent decisions and uncertainties. Decision-making problems (5) and (6) become a problem of robust decision-making if random indicators $f_i(x,\omega)$ and $i = \overline{0,m}$ are represented in the form of non-smooth quantile-based indicators as in (2), reflecting, for example, critical thresholds or imbalance (shortfall or excess), signaling a systemic failure. In the case of discontinuous functions, $f_i(x,\omega)$, expected values, $F_i(x)$, of constraint (6) often represent the risks of different parts ($i = \overline{0,m}$) of the system characterized by probabilistic chance constraints: $P[f_i(x,\omega) \geq 0] \geq 1 - p_i$, $i = \overline{1,m}$. Parameter $p_i$ defines the level of safety or security (see, e.g., References [18–31]) of subsystem $i = \overline{1,m}$. For examples, if $p_i$ is equal to $10^{-3}$, $p_i = 10^{-3}$, it means that supply–demand balances (reserves, thresholds, norms, standards, etc.) can be violated only once in 1000 years. Let us denote a quantile of $f_i(x,\omega)$ by $Q_i(x)$, $i = \overline{0,m}$. Robust version of STO models (5) and (6) can be formulated as follows: maximize

$$Q_0(x) + \mu_0 Emin\{0, f_0(x,\omega) - Q_0(x)\} \tag{7}$$

It is subject to the following constraints:

$$Q_i(x) + \mu_i Emin\{0, f_i(x,\omega) - Q_i(x)\} \geq 0 \tag{8}$$

where $\mu_i > 1$ are parameters regulating the deviation of $f_i(x,\omega)$ below critical quantiles, $Q_i(x)$, and the mathematical expression $Emin\{0, f_i(x,\omega) - Q_i(x)\}$ indicates the expected shortfall. The straightforward use of $Q_i(x)$ destroys concavity of functions, $F_i(x)$. To avoid this, according to model (2) and Equations (3) and (4), the model (7,8) can be equivalently rewritten as Equation (4): maximize w.r.t. $(z,x)$ function

$$z_0 + \mu_0 Emin\{0, f_0(x,\omega) - z_0\} \tag{9}$$

It is subject to the following constraints:

$$z_i + \mu_i Emin\{0, f_i(x,\omega) - z_i\} \geq 0, \; i = \overline{1,m}. \tag{10}$$

The models (9) and (10) are both a concave STO problem if functions $f_i(\cdot,\omega)$ are concave. In the Proposition below, we show that components $z_i^*(x)$, $i = \overline{1,m}$, solvings (9) and (10) w.r.t. $z = (z_0, z_1, \dots, z_m)$ are quantiles $Q_i(x)$. Therefore, models (9) and (10) are robust versions of models (5) and (6), where mean values, $Ef_i$, are substituted by quantiles of indicators, $f_i$, with a security (safety) levels, $\mu_i$, controlling their variability. The models (9) and (10) can also be viewed as concave versions of STO models with probabilistic safety constraints.

**Proposition** (Quantiles of $f_i(x,\omega)$). *Assume $f_i(x,\cdot)$, $i = \overline{0,m}$, have continuous densities, $\mu_i > 1$, $(z^*, x^*)$ is a solution of models (9) and (10) and $\lambda^* = (\lambda_1^*, \dots, \lambda_m^*) \geq 0$ is a dual solution. Then, for $i = 0$ and active constraints, $i = \overline{0,m}$:*

$$P[f_i(x^*,\omega) \leq z_i^*] = 1/\mu_i, \; i = \overline{0,m}. \tag{11}$$

**Proof.** Let $\phi_i(z_i, x, \omega) := z_i + \mu_i min\{0, f_i(x, \omega) - z_i\}$. According to the duality theory it follows that $z_i^*$ maximizes. $\square$

$$E\phi_0(z_0, x^*, \omega) + \sum_{i=1}^{m} \lambda_i^* E\phi_i(z_i, x^*, \omega),$$

where $Emin\{0, f_i(x^*, \omega) - z_i\}$ defines the expected shortfall between $f_i(x^*, \omega)$ and $z_i$, $i = \overline{0, m}$. If $\lambda_i^*$ is positive, $\lambda_i^* > 0$, then $z_i^*$ maximizes $E\phi_i(z_i, x^*, \omega)$, $i = \overline{0, m}$. Therefore, Equation (9) follows from (4) for $i = \overline{1, m}$. From the complementary condition, $\sum_{i=1}^{m} \lambda_i^* E\phi_i(z_i^*, x, \omega) = 0$, and Formula (3) follows Equation (11) for $i = 0$.

Let us also not that the variability of outcomes, $f_i(x, \omega)$, can be controlled by using a vector of quantiles, $z^i = (z_{i0}, z_{i1}, \ldots, z_{il})$, generated as in (9,10), by performance indicators $\sum_l(z_{il} + \mu_{il}min\{0, f_i(x, \omega) - z_{il}\})$, $i = \overline{0, m}$, and $1 < \mu_{i1} < \mu_{i2} < \cdots$. Based on Equation (11), it is possible to say that the models (9) and (10) are defined by VaR (probabilistic chance constraints) and CVaR (expected shortfalls) risk measures (e.g., see Reference [46]) controlling security/safety of the overall system, i.e., a systemic risk. Specifically, the term $\mu_i Emin\{0, f_i(x, \omega) - z_i\}$, $i = \overline{0, m}$, in (9,10), emerging through the problem's transformation with chance constraints (11), is essentially the expected shortfall or CVaR risk measure. Robust quantile-based models (7–10) can always be formulated in terms of the two-stage STO with strategic ex-ante first-stage solutions $x$, complemented with scenario-dependent second-stage solutions to adaptively adjust solutions $x$ to each scenario $\omega$ (see Sections 2.2 and 3). Let us consider a basic two-stage STO model for production planning, enabling us to cope with systemic risks of imbalances in supply–demand relations.

*2.2. Robust Two-Stage Strategic–Adaptive Production Planning Model, Critical Quantiles, and Security Constraints*

This section illustrates how the concept of a two-stage stochastic optimization and robust strategic–adaptive decisions can assist robust decision analysis and system's security management. Consider a basic model, which may correspond to a problem of a decision maker (regional planner) concerned with investments into agricultural production technologies (irrigation and storage), bio-based electricity production facility, water-reservoir infrastructure, or any other long-term and costly project associated with possible irreversibility costs [47]. Assume there is only one producer (region) operating in uncertain environment and this producer needs to satisfy some demand. One of the variables (or both variables) can be stochastic. In this section, let us consider a very simple relation between the production capacity, $x$, and the stochastic demand, $\omega$.

The imbalance between the demand and supply can induce an exceedance of a vital systemic threshold and, as a result, a systemic failure. Planning production, $x$, before exact information on the level of stochastic demand, $\omega$, is available can result in two situations: (a) If the production is lower than the demand, then the producer can potentially fulfill the demand by storage withdrawal ("borrowing") at price $\alpha > c$. If the demand is lower than the production, $x$, the producer can store the excess production at price $\beta < c$. The shortage and the excess production indicate systemic imbalances and risks, which require additional actions and costs for borrowing and storing. In general, costs $\alpha$ and $\beta$ can be interpreted as penalties for not having an adequate capacity, $x$.

Let us denote the shortage or excess of production by $y^+ = \omega - x$ and $y^- = x - \omega$, respectively. The balance equation for a given $\omega$ can be written as $x = \omega + y^+ + y^-$, and the total cost for the production, borrowing, and storing is defined as follows:

$$f(x, \omega) = cx + \begin{cases} \alpha(\omega - x^*) = \alpha y^+, & if \ \omega \geq x \\ \beta(x - \omega) = \beta y^-, & if \ \omega < x, \end{cases} \tag{12}$$

where $\alpha$ is the unit shortage cost and $\beta$ is the unit storage cost (holding cost). The problem is to find the production level of $x^*$ that is optimal (in a sense) for all foreseeable demands, $\omega$, rather than a scenario-specific production $x \rightarrow x(\omega)$, when for each stochastic scenario,

$\omega$, the production level is $x = \omega$. For the production model with cost function (12), the problem of robust planning minimizes the expected costs function:

$$F(x) = E\{f(x, \omega)\} = cx + \alpha \int_{\omega \geq x} (\omega - x) P(d\omega) + \beta \int_{x \geq \omega} (x - \omega) P(d\omega) \qquad (13)$$

This can also be written as a stochastic minimax problem:

$$F(x) = cx + E\{max[\alpha(\omega - x), \beta(x - \omega)]\} = cx + E\{max[\alpha y^+, \beta y^-]\}. \qquad (14)$$

The optimality condition for the optimal first-stage production level, $x > 0$, $F'(x) = 0$, yields the following equation:

$$P[\omega \leq x] = \frac{\alpha - c}{\alpha + \beta}. \qquad (15)$$

The optimal production level, $x_p^*$, is the quantile $\omega_p$, which satisfies the safety constraint (15) for $p = \frac{\alpha - c}{\alpha + \beta}$. From (15), the probability that the production is higher than the demand is defined by the four main parameters: the probability distribution of the demand, the production (investment) costs, $c$, and the operational costs $\alpha$ and $\beta$ associated with managing scenario-specific production shortages or surpluses, $y^+$ and $y^-$. The optimal second stage solution $y^*(x^*, \omega)$ is defined as $y^*(x^*, \omega) = max[\alpha(\omega - x), \beta(x^* - \omega)]$. Equation (15) means that the level of production $x^*$ will satisfy or even be greater than stochastic demand at a given probability, $p$, defined by the investment costs and the operational costs associated with adaptive actions towards managing production excess and shortage. Although not explicitly, risk aversion is introduced in the model through the second-stage decisions. Thus, strategic decisions, $x^*$, cover only a slice of demand $\omega$ while the rest is a adapted ex-post. Therefore, the solution of the two-stage strategic–adaptive stochastic problem fulfills the production security level. The indicator $P[\omega \leq x]$ defines the Value-at-Risk (VaR), i.e., the probability and the critical quantile of $\omega$, which cannot be fulfilled by strategic decisions.

## 3. Two-Stage Strategic–Adaptive GLOBIOM Model

Section 2.2 illustrates the concept of a robust two-stage strategic–adaptive decision-making with a basic model of a single producer (region, system) facing risks, associated with production–supply imbalances. In the example, regional systemic security depends on the regional feasible decisions $x$ and $y$, the representation (e.g., by means of a probability distribution or probabilistic scenarios) of the stochastic demand, and investment and operational costs.

In this section, we formulate a two-stage strategic–adaptive large-scale, multi-regional, multi-sectoral GLOBIOM model [33], using similar quantile-based robustness principles as in (3,4). In GLOBIOM, the key variables, i.e., costs, prices, production portfolio, distribution of systemic shocks, etc., depend endogenously on complex supply–demand chains included in the model. Systemic disturbances in GLOBIOM are triggered by different scenarios $\omega$ of potential crop yield shocks.

### 3.1. Two-Regional GLOBIOM: Systemic Risks, Risk Exposure, and Risk Sharing

Before formulating the general two-stage strategic–adaptive GLOBIOM model, let us use its two-regional (two-sectorial) fragment to discuss the issues of potential systemic imbalances, cooperation, and risk sharing. We aim to demonstrate that systemic risks emerge as a result of systems' interactions, joint resource and security constraints, risks exposures, and decisions of various agents. In this case, a risk-free region (not exposed to exogenous shocks) can become risk-exposed through joint constraints and goals. For illustrative purposes, the demand, $d$, is assumed to be deterministic. In the general model, the demand is stochastic and depends on shocks and demand-price elasticities. Production capacities of regions are defined by $x_i$ and $c_i$ are production costs, $I = 1,2$. Regional production technologies include a backup technology $y \geq 0$ characterizing a potential

production shortfall. Price $b$ of technology $y$ can be interpreted as a cost (or penalty) of not having sufficient capacities $x_i$, $I = 1,2$, to fulfill the demand. Assume $c_1 < c_2 < b$, i.e., production in the first region is the cheapest. There is no production depletion due to uncertainties, $a_1, a_2 = 1$. The goal of a region without uncertainties is formulated as to find such $x_i$ that minimize the cost function:

$$c_1 x_1 + c_2 x_2 + by \tag{16}$$

S.t.

$$a_1 x_1 + a_2 x_2 + y > d \tag{17}$$

where capacities $x_1$ and $x_2$ have to fulfill the joint demand constraint (17). The optimal solution of this deterministic problem is $x_1^* = d$, $x_2^* = 0$, $y^* = 0$, i.e., only the most efficient region/sector/activity is involved in production.

In the presence of uncertainty, the level of production (outputs) $x_1$, $x_2$ of both regions can be reduced (dependently or independently) by distortions $a_1, a_2$, $0 \leq a_i < 1$, $i = 1,2$, e.g., yield shocks, (in)efficiencies, and resource shortage (biomass, water, and solar or wind energy). Systemic imbalances (risks) are induced through joint supply–demand constraints. Similar to Section 2.2., production shortfall can be potentially covered by the adaptive ex-post scenario-dependent decisions $y(\omega) \geq 0$, e.g., storage withdrawals or imports. The variability of $y(\omega)$ defines the level of production and food security.

The robust two-stage strategic–adaptive problem is formulated as a problem of minimization of the total expected cost:

$$c_1 x_1 + c_2 x_2 + bEy(\omega) \tag{18}$$

It is subject to production security constraints, as follows:

$$a_1(\omega)x_1 + a_2(\omega)x_2 + y(\omega) \geq d \tag{19}$$

For all scenarios $\omega$. If supply $a_1(\omega)x_1 + a_2(\omega)x_2$ is less than demand $d$, the difference $y(x, \omega) = d - a_1(\omega)x_1 - a_2(\omega)x_2$ indicates the supply shortage. The constraint (19) introduces path-dependencies, interactions, and trade-offs among the strategic and the adaptive decisions. The problems (18) and (19) can be formulated as a non-smooth stochastic minimax problem: minimize

$$F(x) = c_1 x_1 + c_2 x_2 + bE max\{0, d - a_1(\omega)x_1 - a_2(\omega)x_2\}$$

where $bE max\{0, d - a_1(\omega)x_1 - a_2(\omega)x_2\}$ is the expected shortfall cost when the demand, $d$, is greater than random supply $a_1(\omega)x_1 + a_2(\omega)x_2$.

Robust strategic decisions, $x_1^*$ and $x_2^*$, satisfy quantile-based risks constraints defined by stochastic variables $a_1(\omega)$ and $a_2(\omega)$, decisions $x = (x_1, x_2)$, security constraints (19), and the costs functions. To illustrate this, assume that $a_1(\omega) < 1$ and $a_2 = 1$, i.e., only first region experiences stochastic shocks and production depletion. In the presence of shocks, $a_i(\omega)$, the less efficient second region ($c_1 < c_2$) can become the key player fulfilling the constraint (19). Assuming that function $F(x)$ has continuous derivatives, the optimal positive solutions $x_1^* > 0$ and $x_2^* > 0$ can be found when $F_{x_1}(0,0) = c_1 - bEa_1(\omega) < 0$ and $F_{x_2}(0,0) = c_2 - b < 0$. If $c_1 - bEa_1(\omega) > 0$, region 1 will not participate in the production. The security requirement (19) will be managed only by region 2. Both regions will stay active only if $c_1 - bEa_1(\omega) < 0$, $Ea_1(\omega)$ defines the expected value of $a_1(\omega)$. It is interesting to note that region 2 always participates in production because $c_2 - b < 0$. Its production level, $x_2^* > 0$, is defined by the following probabilistic equation:

$$P[d - a_1(\omega)x_1^* \geq x_2^*] = c_2/b \tag{20}$$

From Equation (20), it follows that the optimal production level, $x_2^*$, of the second risk-free region is defined by the quantile of the probability distribution, characterizing

shocks, $a_1(\omega)$, in region 1. The strategic ex-ante scenario-independent decisions, $x^*$, can manage only a part of risks, defined by the ratio of costs, $c_2/b$. The other part of the risks, i.e., the production shortage, has to be managed by adaptive ex-post scenario-dependent decisions. Thus, the risk-free region 2 becomes exposed to systemic risks characterized by the interdependencies between the demand ($d$), shocks ($a_i(\omega)$), cost functions ($c_i$ and $b$), robust decisions ($x_1^*$ and $x_2^*$), and the security constraint (19). These risks can be regulated by parameters $c_1$ and $c_2$ on local (regional) levels, and $b$ and $d$ on global (national and international) levels. Equations of type (20) are known as chance (or probabilistic) constraints [48], safety or reliability constraints [17], or Value-at-Risk (VaR) constraints [18–22]. Similar probabilistic equations are solved to manage systemic risks in a large-scale multi-regional, multi-sectorial GLOBIOM in Section 3.2.

### 3.2. Robust Systemic Risks Management in a Two-Stage Strategic–Adaptive GLOBIOM

The supply-and-demand relations in the general GLOBIOM are represented by much more complex supply–demand chains than in the two-regional model of Section 3.1. GLOBIOM is a multi-regional, multi-sectorial recursive-dynamic, partial-equilibrium model designed to address various land-use-related topics, such as bioenergy production potentials and policy impacts, sustainable agricultural production and climate change adaptation, implications of deforestation, etc. The market equilibrium is solved by maximizing the sum of producer and consumer surplus subject to resource, technological, political, and FEWE security constraints. The demand for high-value, high-quality commodities like crops, fruits, vegetables, livestock products, fisheries, and oils is growing with increasing population and changing consumption preferences. Producers in different regions and management systems (intensive, mixed, and traditional) are trying to diversify and adjust their production portfolios and management technologies accordingly. The model accounts for about 18 most important crops, a range of livestock production activities, forestry commodities, first- and second-generation bioenergy, and water. GLOBIOM endogenously calculates demand and supply quantities, bilateral trade flows, and prices for commodities and natural resources at 10-year-step intervals, from 2000 up to 2050.

The model includes about 37 major world regions. Each region is represented by a set of feasible agricultural, bioenergy, and forestry sectors management practices; production and processing technologies; supply chains; costs; efficiencies; production and market uncertainty; and risks. Region-specific goals and constraints allow for policy analysis of land-use competition and land-use changes driven by increasing demand and changing consumption preferences for food, feed, water, and biofuels [49–54], at global and regional levels. The main land uses comprise agricultural land, grassland, natural and managed forest, fast-rotation forest plantations, and natural land. This gives planners a basis for the analysis of future land-use options and for identifying possible shortfalls in food, feed, and biomass production/supplies.

The model incorporates FEWE security constraints. The food security constraint defines the necessary energy intake (in kilocalories per capita), feeds security constraint requires that feeds produced for livestock from crops, grass, and byproducts confirm the livestock dietary requirements in energy (in megacalories per livestock unit). Biofuel production targets, i.e., biofuels (energy) security constraint, have to be fulfilled from crops, woody biomass, and agricultural residues, i.e., the joint constraint on the biofuels production from the first- and the second-generation biofuels. Because of uncertain production, the joint food/feed/biofuel security constraints introduce competition for the natural resources (land and water) and the trade-offs between the allocation of the resources to resource-based sectors, producers, and consumers. Land-based agriculture and forestry sectors depend on each other through joint bioenergy targets [33,34], which can induce systemic risks similar to the model in Section 3.1. The energy biomass can be converted through various processes characterized, in general, by uncertain transfer and conversion coefficients, resource utilization, etc. A formal description of GLOBIOM model can be found in References [33,34].

The goal function of the two-stage strategic–adaptive GLOBIOM model is to maximize the total expected producer and consumer surpluses (21) with respect to decision variables $(x, y(\omega))$ subject to food, energy, water, and environmental security constraints (22):

$$F(x) = E_\omega f(x, y(\omega), \omega) = \int f(x, y(\omega), \omega) P(d\omega) \tag{21}$$

It is under the following constraints:

$$g_i(x, y(\omega), \omega) \leq 0, \ i = \overline{1, m}. \tag{22}$$

Uncertainty, $\omega$, can be represented by a finite set of probabilistic scenarios, $\omega^s$, $s = \overline{1, S}$, which allows us to reformulate models (21) and (22) as the problem of maximizing goal function:

$$\sum_{s=1}^{S} p_s f(x, y^s, \omega^s) \tag{23}$$

It is subject to the following constraints:

$$g_i(x, y^s, \omega^s) \leq 0, \ i = \overline{1, m}, \ s = \overline{1, S}. \tag{24}$$

Scenarios $\omega^1$, $\omega^2$, ..., $\omega^S$ and respective probabilities $p_1$, ..., $p_S$, $\sum_{s=1}^{S} p_s = 1$ can be derived from historical observations, from simulation models (e.g., EPIC model [54]), or/and expert opinions. If information is not available, it is possible to assume that $p_s$, $s = \overline{1, S}$, are uniformly distributed, $p_s = 1/S$.

Constraints (24) can be fulfilled only for some scenarios. Let us introduce auxiliary decision variables $z_{is} \geq 0$, fulfilling equations:

$$g_i(x, y^s, \omega^s) \leq z_{is} \tag{25}$$

Variable $z_{is}$ identify the "lack" of security. It can be viewed as systemic imbalances (shortages and shortfalls) at regional (and global, $\sum z_{is}$) levels, requiring additional actions regarding introduction of more advanced production chains, production technologies, investments, and insurance mechanisms. Thus, the adaptive decisions, $y^s$, can be viewed as already existing measures and actions (adaptive decisions), whereas decisions $z_s$ identify the "demand" for additional technologies and measures that must be implemented to ensure a required regional and global systemic security level. Introducing the new capacity $z_s$, we consider the problem of maximizing function:

$$\sum_{s=1}^{S} p_s f(x, y^s, \omega^s) - \sum_{s=1}^{S} p_s(\pi_s, z_s), \tag{26}$$

with respect to $x$, $y^s$, and $z_s = (z_{1s}, \ldots, z_{ms})$, subject to security constraints (25), where $\pi_s = (\pi_{1s}, \ldots, \pi_{ms}) \geq 0$ are vectors of weights or penalties, defining losses associated with insecurity, costs of threshold exceedance, etc.; $(\cdot, \cdot)$ denotes a scalar product of vectors. If optimal solutions $x^*$, $y_s^*$, and $z_s^*$ of models (25) and (26) have some $z_s^* \neq 0$, then Equation (24) is not satisfied for all scenarios. Security level of the models (25) and (26) can be regulated by parameter $\pi_s$. A larger $\pi_s$ imposes stronger security requirements. Similar to model in Section 3.1., the security requirements can have the form of probabilistic constraints ([18,22–26,29–31]):

$$P[g_i(x, y(\omega), \omega) - g_i^* \geq 0] \geq \gamma_i \tag{27}$$

where $g_i^*$ is a targeted level of indicator $g_i(x, y(\omega), \omega)$, for example, food or feed requirements, environmental norms, and bioenergy mandates. Parameter $\gamma_i$ regulates the security level specifying the desirable probability of fulfilling the constraint (27). The probabilistic constraints are often substituted (see, e.g., References [19,22,24,25]) by the expected shortfall functions:

$$E\max\{0, g_i(x, y(\omega), \omega) - g_i^*\} \tag{28}$$

Vector $z_s = (z_{1s}, \ldots, z_{ms})$ in (26) can be defined explicitly as $z_{is} = max\{0, g_i(x, y^s, \omega^s) - g_i^*\}$ and the problem (25,26) is reformulated as maximizing:

$$\sum_{s=1}^{S} p_s f(x, y^s, \omega^s) - \sum_{i=1}^{m} \sum_{s=1}^{S} p_s \pi_{is} max\{0, g_i(x, y^s, \omega^s) - g_i^*\}, \tag{29}$$

or equivalently as maximizing function (26), as in the following:

$$\sum_{s=1}^{S} p_s f(x, y^s, \omega^s) - \sum_{s=1}^{S} p_s(\pi_s, z_s) \tag{30}$$

Under constraints:

$$z_{is} \geq g_i(x, y^s, \omega^s) - g_i^*, \; z_{is} \geq 0, \; i = \overline{1, m}, \; s = \overline{1, S}. \tag{31}$$

The global systemic risks in the general multi-regional multi-sectorial GLOBIOM model are related to critical imbalances and violation of security constraints (24,25) regulated by vector $z_s$ and parameter $\pi_s$ in (26). Random variable $\pi_s z_s^*$ measures the risks of violation security constraint (24), whereas the random sum $\sum \pi_s z_s^*$ characterizes total (global) systemic risk.

Let us consider a general formulation of the goal function (29):

$$F(x) = E\left[ f(x, y(\omega), \omega) - \sum_{i=1}^{m} \pi_i(\omega) max\{0, g_i(x, y(\omega)) - g_i^*\} \right]. \tag{32}$$

Assuming that $F(x)$ has continuous partial derivative $F_{x_j}(x^*)$, we obtain the optimal condition of systemic risk equilibrium: if $x_j^* > 0$, then

$$F_{x_j}(x^*) = c_j - \sum_{i=1}^{m} \pi_i a_{ij} P[g_i(x, y(\omega), \omega) \geq g_i^*] = 0. \tag{33}$$

Thus, global systemic risk indicators $P[g_i(x, y(\omega), \omega) \geq g_i^*]$ and the corresponding robust complementary strategic ex-ante and adaptive ex-post solutions $(x, y(\omega), z(\omega))$ are derived by solving a system of linear Equation (33) with respect to these indicators (similar to the two-regional probabilistic Equation (20)). It is not possible to derive these indicators analytically. Therefore, in the next section, we discuss selected numerical results, in particular, histograms of global storage withdrawals $\sum z_{is}$ hedging regional and global production shortfalls and systemic risks by relaxing tight supply–demand relations in a complex multi-regional multi-sectorial GLOBIOM model.

## 4. Selected Numerical Results: Strategic and Adaptive Decisions

In the two-stage strategic–adaptive GLOBIOM, land allocation to different land-use systems is a strategic ex-ante decision. Other strategic decisions are water and grain storage capacities. Decisions regarding bilateral trade flows, and water and grain storage withdrawals are operational-scenario-dependent decisions. They help strategic decisions to adjust to each shock scenario.

Selected numerical experiments demonstrate the differences between the two alternative cases of GLOBIOM-based analysis. In the first case (C1), the GLOBIOM model generates results corresponding to various crop yield scenarios. This means that approach C1 calculates scenario-dependent decisions regarding crop and technological portfolios, which can be quite different for different yield scenarios. A scenario-by-scenario approach does not incorporate trade-offs between strategic and adaptive decisions, as was discussed in Section 3. Practical implementation of scenario-dependent decisions can lead to high reversibility costs [39] and magnify systemic imbalances and risks, if an other-than-expected uncertainty scenario occurs.

The second case (C2) involves the two-stage strategic–adaptive GLOBIOM for robust planning under uncertainty and inherent risks. In C2, land allocation between land-use systems corresponds to strategic ex-ante scenario-independent decisions. The strategic decisions are "corrected" by adaptive ex-post scenario-specific decisions (trade, storage

withdrawals, and prices) after the information about the actual scenario becomes available. The combination of the interdependent strategic and adaptive decisions ensures the systems flexibility and preparedness through robust coordination of systems responses to risks of various kinds. The robust combination of the strategic and adaptive decisions minimizes the total costs associated with both types of the decisions, in particular, with adaptive operational actions in response to emerging systemic disturbances, triggered by each crop yield shock. By varying parameter $\pi$ in (26), and depending on endogenously calculated trade costs, it is possible to investigate regional potential for the required level of regional self-sufficiency, i.e., region-specific reliance on domestic production and storage vs. international trade.

Robust storages calculated with C2 introduce flexibility in supply–demand relations. Storages are important to avoid large and often irreversible investments into strategic decisions, e.g., for expansion of agricultural land or irrigation systems. Figure 1 presents histograms of storage withdrawals for selected grain and oilseed commodities.

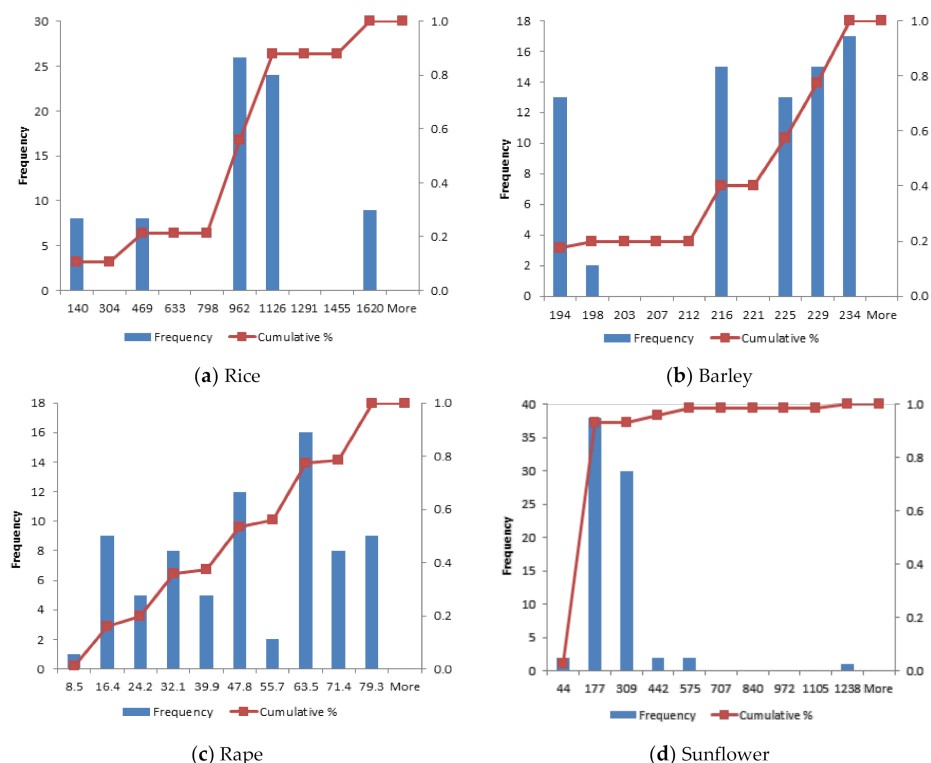

**Figure 1.** Distribution of global storage withdrawals for selected crops ((**a**) rice, (**b**) barley, (**c**) rape, and (**d**) sunflower). Withdrawals characterize global systemic risks of supply–demand imbalances. "Frequency" on the vertical axis defines the absolute number of withdrawals; the value of withdrawals is defined on the horizontal axis (in thousand tons). The percentage of total withdrawals at or below the value on the horizontal axis is defined as "cumulative".

Robust reserves (Figure 1) are within reasonable ranges, if compared to global grains production [55]. It is worth mentioning that rape and sunflower seeds are highly demanded for biofuel production. Strict biofuel targets and other possible constraints, such as (24) or (25), increase prices of all agricultural commodities used for direct and indirect consumption. Relaxing biofuel obligations by the amount equivalent to the rape and sunflower reserves in Figure 1 can considerably decrease the prices for all agricultural commodities. Contrary to the historical disequilibrium stock-to-use ratio, stochastic GLOBIOM can evaluate robust reserves necessary to reduce the prices to acceptable levels.

Figure 2 presents the percentage of total land allocated to different land-use systems, calculated with the two-stage strategic–adaptive GLOBIOM (C2): if GLOBIOM is run

with average yield scenario (average yield scenario, C1) and if GLOBIOM is run with extreme shock scenario (2000 yield shock scenario, C1). In C2, the robust interdependent strategic and adaptive decisions minimize the reversibility costs, and, therefore, the C2 settings recommend more sustainable actions. For example, the decisions in C2 approach focus on natural ecosystems preservation more than in the C1 approach. It is suggested to slow down the increase of managed forests resulting from natural forest conversion. The grassland, as an important source of feeds, should also be saved from conversion (see panels b and c in Figure 2). If needed, all land conversions can occur within suitable natural land (Figure 2, panel f). Instead of dedicating agricultural land, biofuels can be produced and biofuel mandates fulfilled through increasing the share of planted forest (Figure 2, panel e).

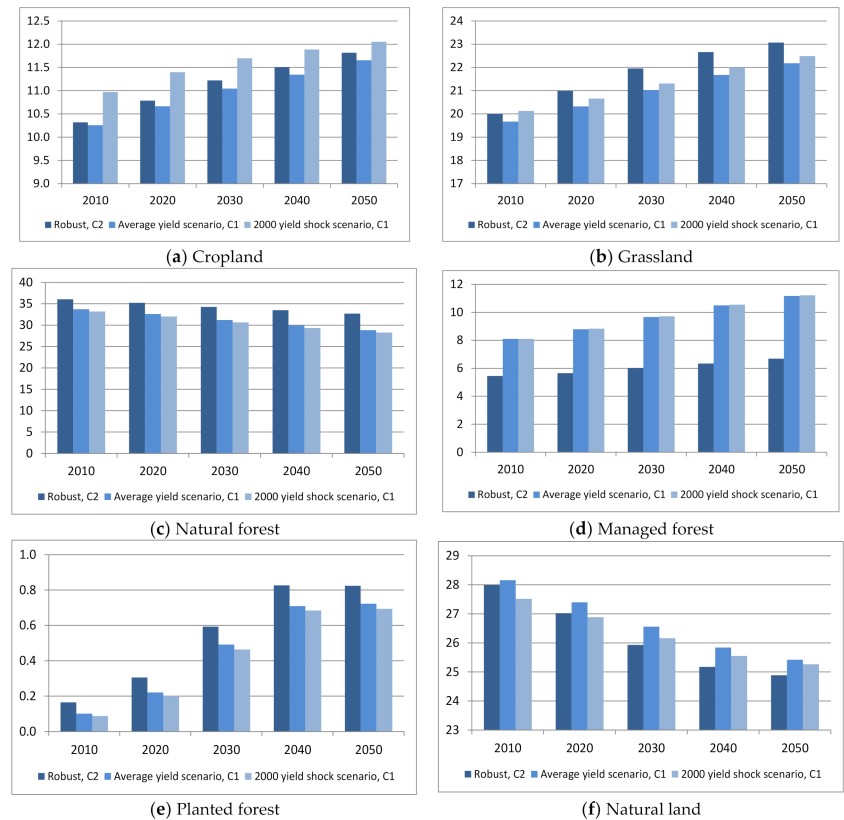

**Figure 2.** Land in different land-use systems, in percentage terms, corresponding to the results of two-stage strategic–adaptive Global Biosphere Management (GLOBIOM) model (C2); if GLOBIOM is run with average yield scenario (average yield scenario, C1); if GLOBIOM is run with extreme shock scenario (2000 yield shock scenario, C1). Simulation year is identified on the horizontal axis and the percentage is on the vertical axis.

## 5. Concluding Remarks

The main issue discussed is that growing interdependencies between FEWE systems increase their vulnerability, especially if the systems are governed by incoherent policies. Any insignificant-at-the-first-glance-shock, e.g., due to weather variability or inadequate, say, biofuel, policies may induce systemic risks that implicitly depend on the whole structure of the systems, in particular, costs, technologies, prices, trade flows, risk exposure, security constraints and targets, risk measures, and decisions of agents. The paper discussed the two-stage STO approach for robust strategic–adaptive management of systemic risks associated with critical imbalances and vital threshold exceedances, affecting FEWE security in complex interdependent systems. We discussed the notion of robust quantile-based estimation in statistics and establish the connection between the robust estimation

and the two-stage non-smooth STO for robust strategic–adaptive decision-making in the presence of uncertainty and risks. To illustrate the emergence of systemic risks, we formulated a basic two-stage production planning model with the two types of decisions (strategic and adaptive), to minimize the risks of systemic imbalances in a supply–demand relation. A two-regional (two-sectorial) version GLOBIOM model illustrates that systemic risks are induced by complex systemic interdependencies, including the distribution of risks shaped by decisions and by the security constraints. In the two-stage STO model, the risk aversion is induced by the coexisting interdependent strategic ex-ante and adaptive ex-post decisions, in the form of quantile-based probabilistic security equations. The general multi-regional, multi-sectorial two-stage strategic–adaptive GLOBIOM model enables us to manage systemic risks by implicitly solving the multidimensional probabilistic security equation, Equation (33). Strategic decisions comprise land allocation by different land uses, land-management technologies, and storage capacities. Adaptive (operational) decisions are trading, demand and price adjustments, and storage withdrawals. Selected numerical results illustrate that the exposure to production and market risks motivates the reliance on domestic grain storages to buffer production shortfalls and meet security requirements at regional and global levels. The two-stage strategic–adaptive GLOBIOM reveals tight supply–demand relations in the systems and identifies robust measures, to relax them, especially in the presence of systemic risks, when trade partners are isolated or tightened by trade bans, and strict constraints and regulations, e.g., on biofuel mandates or green deal policies. The results demonstrate that robust storages (grain and water reserves) may prevent from unwanted land expansion and from undertaking costly investments, e.g., in additional irrigation technologies. The two-stage strategic–adaptive GLOBIOM derives a robust combination of preventive strategic and adaptive operational decisions, which allow for qualitatively different policy recommendations, if compared to a traditional scenario-by-scenario decision-making analysis.

**Author Contributions:** All authors contributed to the developments of methodology, software, and interpretation of results. All authors have read and agreed to the published version of the manuscript.

**Funding:** This research received no external funding.

**Institutional Review Board Statement:** Not applicable.

**Informed Consent Statement:** Not applicable.

**Data Availability Statement:** Not applicable

**Conflicts of Interest:** The authors declare no conflict of interest.

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
