# Peer review of "Robust Management of Systemic Risks and Food-Water-Energy-Environmental Security: Two-Stage Strategic-Adaptive GLOBIOM Model"

_sustainability, doi:10.3390/su13020857_

Round 1
Reviewer 1 Report
See attached file.

Author Response
Answers to a reviewer comments on the paper “Robust management of systemic risks and food-water-energy-environmental security: two-stage strategic-adaptive GLOBIOM model”, Tatiana Ermolieva, Petr Havlik, Yurii Ermoliev, Nikolay Khabarov, Michael Obersteiner
We are thankful for the important comments, which allowed to improve our paper. In what follows, we present the detailed answers to the reviewer’s comments.
General comments:
This paper considers the problem of optimal decision making with quantile-based constraints. The problem under study includes a short-term decision which induces additional long-term costs from under-supply of a specific good. Risk is included in this system through random future outcomes for, e.g., production shortfalls. The authors conclude the work with numerical case studies demonstrating the decisions that would be made under different scenarios. My primary concern is that this work does not make clear its own novelty. Mathematically, the formulation and algorithms are all straight-forward and do not seem to include any innovations. Additionally, the problem of interest (GLOBIOM) has also been studied extensively in the past. The authors need to emphasize their innovations and motivate why this approach is superior to prior methods....
Reply: Thank you for the comment. Of course, the two-stage STO has been used in the past. However, the two-stage nonsmooth STO with probabilistic security/safety/chance constraints is still not very widely used for decision-making and especially - for robust decision making. We clarify what is meant under robust decision making and establish the connection between the robust estimation and the robust two-stage nonsmooth STO.
More specifically, in this paper we argue that the two-stage STO is extremely important and novel approach for managing systemic interdependent risks in interacting FEWE systems. The robust combination of strategic and adaptive decisions provides the systems with flexibility and robustness to effectively manage the imbalances, which otherwise can trigger systemic failures.
We introduce the definition of systemic risks in interdependent systems as “Systemic risks in interdependent FEWE systems can be defined as the risks of a subsystem (a part of the system) threatening the sustainable performance and achievement of FEWE security goals. Thus, a shock in a peripheral subsystem induced (intentionally or unintentionally) by an endogenous or exogenous event, can trigger systemic risks propagation with impacts, i.e. instability or even a collapse, at various levels….”
We discuss the notion of robustness in statistics and make an explicit connection between the robust quantile-based nonsmooth estimation problem in statistics and the two-stage nonsmooth STO problem with probabilistic security/safety/chance constraints of robust strategic-adaptive decision making. Then we clarify that robust management of systemic risks in interdependent systems under FEWE probabilistic security/safety/chance constraints can be achieved “… by solving a system of probabilistic security equations”.
We introduced additional verbal description of GLOBIOM model, however, we preferred to preserve a rather general formulation of the model. Introduction of additional detailed equations and constraints would significantly divert the attention of the readers from the main message that the robust management of systemic risks can be achieved “… by solving a system of probabilistic security equations” within a framework of a two-stage nonsmooth STO problem with probabilistic security/safety/chance constraints of robust strategic-adaptive decision making. There are several places in the paper where we mention this, e.g. sentence in line 99 of the revised version “… The imbalances are represented by means of expected surpluses and shortfalls (deficits) defining the systemic risks and characterizing the demand for additional strategic and operational measures to reduce or eliminate the imbalances. Management of systemic imbalances can be addressed by solving a system of probabilistic security equations as it is discussed in this paper (sections 2, 3, 4). ”
These and other issues are also clarified below in the Detailed comments.
Detailed comments.
. Please review this paper for typos and grammatical errors. There are a few throughout this work.
Reply: We improved the grammatical errors and typos.
. Please introduce the notation that is used in this work somewhere….
Reply: We introduced the missing notation, e.g. in line 156 of the revised document: “… The probability space () is defined by stochastic parameters , , event space and the probability measure or probability distribution .” We also introduced missing notation, e.g. throughout the paper.
. The introduction presents this paper as being part of a larger literature on systemic risk. However, the definition of systemic risk in the power grid (see, e.g., Cassidy, A., Feinstein, Z., &Nehorai, A. (2016). “Risk measures for power failures in transmission systems." Chaos: An Interdisciplinary Journal of Nonlinear Science, 26(11), 113110), finance (see, e.g., Glasserman, P., & Young, H. P. (2015). “How likely is contagion in financial networks?." Journal of Banking & Finance, 50, 383-399.), etc. all consider different notions of systemic risk. Please place your definition of systemic risk within the literature more clearly; this includes providing a larger literature review.
Reply: Thank you very much for providing us with the references. We included them in the document. We introduced the following sentence in line 53 defining the systemic risks in interdependent FEWE system:
“Systemic risks in interdependent FEWE systems can be defined as the risks of a subsystem (a part of the system) threatening the sustainable performance and achievement of FEWE security goals. Thus, a shock in a peripheral subsystem induced (intentionally or unintentionally) by an endogenous or exogenous event, can trigger systemic risks propagation with impacts, i.e. instability or even a collapse, at various levels.”
Also, we added a sentence in line 99 of the revised version:
“… The imbalances are represented by means of expected surpluses and shortfalls (deficits) defining the systemic risks and characterizing the demand for additional strategic and operational measures to reduce or eliminate the imbalances. Management of systemic imbalances can be addressed by solving a system of probabilistic security equations as it is discussed in this paper (sections 2, 3, 4). ”
. Line 50-51 states, “These are examples of systemic risks that are analytically intractable and cannot be characterized by a probability distribution." Such a statement is misleading; probability distributions can be simulated (for instance) through Monte Carlo simulations.
Reply: We excluded this sentence in the revised version of the paper.
. Line 54 states, “a combination of exogenous shocks and incoherent decisions of intelligent agents." Is the “incoherence" of the decision important? One key result from game theory is that each agent can act rationally and still result in suboptimal outcomes. Please clarify.
Reply: We excluded this sentence in the revised version of the paper. We omit “coherent” as it can lead to misunderstandings. In relation to systemic risks, we mention that “These (systemic) risks are shaped by systemic interactions, risk exposures and decisions of various agents.”
. Line 162 (and elsewhere): Please consider cleaning up notation.
For the detailed answers with formulas and notation, please see the attached comments and answers to the reviewers' report

Reviewer 2 Report
The paper proposed for publication was structured in interesting and rigorous way. The introduction is well written and perfectly explains the objectives to be pursued. The methodology is complex but clearly explained. The results may have important scientific resonance.
I find no changes to indicate to the authors but I suggest a thorough English revision.
Author Response
Answers to a reviewer comments on the paper “Robust management of systemic risks and food-water-energy-environmental security: two-stage strategic-adaptive GLOBIOM model”, Tatiana Ermolieva, Petr Havlik, Yurii Ermoliev, Nikolay Khabarov, Michael Obersteiner
We are thankful to the reviewer for the high opinion about our contribution to the Sustainability journal. We improved typos, introduced missing notation, introduced additional clarifications to our approach based on two-stage nonsmooth STO under safety/security/chance constraints for managing systemic interdependent risks in interacting food, energy, water, environmental systems.
Round 2
Reviewer 1 Report
The new version is a significant improvement over the original paper. I only have a minor comment that "Prob" as written on (for instance) line 162, should be the introduced probability measure "P".
Author Response
We are thankful to our reviewer for the comments and suggestions.
Reviewer's comment: "The new version is a significant improvement over the original paper. I only have a minor comment that "Prob" as written on (for instance) line 162, should be the introduced probability measure "P"."
Reply: We improved the notation (Prob) in line 162 and in similar places in the text, changed "Prob" to "P".